# Clinical Significance of Adverse Events for Patients with Unresectable Hepatocellular Carcinoma Treated with Lenvatinib: A Multicenter Retrospective Study

**DOI:** 10.3390/cancers12071867

**Published:** 2020-07-11

**Authors:** Shigeo Shimose, Hideki Iwamoto, Takashi Niizeki, Tomotake Shirono, Yu Noda, Naoki Kamachi, Shusuke Okamura, Masahito Nakano, Hideya Suga, Ryoko Kuromatsu, Taizo Yamaguchi, Takumi Kawaguchi, Masatoshi Tanaka, Kazunori Noguchi, Hironori Koga, Takuji Torimura

**Affiliations:** 1Division of Gastroenterology, Department of Medicine, Kurume University School of Medicine, Kurume 830-0011, Japan; niizeki_takashi@kurume-u.ac.jp (T.N.); shirono_tomotake@med.kurume-u.ac.jp (T.S.); noda_yuu@med.kurume-u.ac.jp (Y.N.); kamachi_naoki@med.kurume-u.ac.jp (N.K.); okamura_shyuusuke@kurume-u.ac.jp (S.O.); nakano_masahito@kurume-u.ac.jp (M.N.); ryoko@med.kurme-u.ac.jp (R.K.); takumi@med.kurume-u.ac.jp (T.K.); hirokoga@med.kurume-u.ac.jp (H.K.); tori@med.kurume-u.ac.jp (T.T.); 2Iwamoto Internal Medical Clinic, Kitakyusyu 802-0832, Japan; ttttyama2@yahoo.co.jp; 3Department of Gastroenterology and Hepatology, Yanagawa Hospital, Fukuoka 832-0077, Japan; suga516@med.kurume-u.ac.jp; 4Department of Gastroenterology and Hepatology, Miyama, Fukuoka 839-0295, Japan; mazzo6528@me.com; 5Department of Gastroenterology and Hepatology, Omuta City Hospital, Fukuoka 836-8567, Japan; hisyo@ghp.omuta.fukuoka.jp

**Keywords:** lenvatinib, adverse events, molecular target agents, hepatocellular carcinoma

## Abstract

We sought to investigate the clinical profile(s) associated with the discontinuation of lenvatinib (LEN) due to severe adverse events (DLSAE) in patients with unresectable hepatocellular carcinoma (HCC). This retrospective study enrolled 177 patients with HCC treated with LEN. Independent factors associated with DLSAE were advanced age, albumin-bilirubin (ALBI) grade 2, fatigue grade ≥ 3, and appetite loss ≥ 2. The overall survival (OS) in the group that did not require DLSAE was significantly longer compared to the group that did require DLSAE (median survival time (MST): not reached vs. 12.8 months, *p* < 0.001). Moreover, advanced age was the most important variable for DLSAE in a decision tree analysis. Hypertension and hand-foot-skin-reaction (HFSR) were also significantly associated with longer survival, and the occurrence of hypertension was the earliest predictor for improved prognosis, while appetite loss and development of grade ≥ 3 fatigue were predictive of a poor prognosis. We concluded that the appearance of hypertension has potential as an early surrogate marker to predict improved prognosis. Moreover, careful management to avoid discontinuation of treatment leads to longer survival in patients receiving LEN.

## 1. Introduction

Hepatocellular carcinoma (HCC) is the most common primary hepatic malignancy and represents a growing cause of public health issues worldwide [1,2,3]. The Barcelona Clinic Liver Cancer (BCLC) staging system is the most widely used algorithm for determining the course of treatment of HCC [4]. Although antiviral therapy contributed to improve prediction of HCC risk [5] patients are often diagnosed at an advanced stage [6]. Molecular-targeted agents (MTAs) are recommended for the treatment of patients with intermediate or advanced stage HCC [7,8] and have increasingly been used as the standard therapy for this patient population.

Lenvatinib (LEN) is an MTA that has been approved as a 1st line treatment for patients with unresectable HCC in the USA, European Union, Japan, and China on the basis of the results of the REFLECT study, a global multicenter randomized phase 3 trial [9]. In this study, LEN demonstrated noninferiority to sorafenib (SORA) with respect to overall survival (OS). It has previously been reported that albumin-bilirubin (ALBI) grade [10,11], neutrophil-to-lymphocyte ratio (NLR) [12], controlling nutritional status (CONUT) [13], and relative dose intensity [14] were predictive factors for OS or therapeutic effect in LEN treatment for HCC. Despite the reported efficacy of MTAs, including LEN, the high rate of adverse events (AEs) currently limits their application in a clinical setting [15]. LEN targets vascular endothelial growth factor (VEGF) receptors 1-3 (VEGFR1-3) and fibroblast growth factor (FGF) receptors 1-4 (FGFR1-4) [16,17], which strongly inhibit tumor angiogenesis. However, recent studies revealed that antiangiogenic drugs, such as LEN, not only inhibited tumor angiogenesis but also inhibited the vascular structure of other healthy organs, such as the thyroid, adrenal grands, and others [18,19].

It has been reported that over 90% of patients treated with SORA or LEN developed AEs [9], and that the occurrence of severe AEs necessitated dose reduction or discontinuation of treatment. Furthermore, discontinuation of LEN due to severe AEs (DLSAEs) was related to the duration of treatment [13]. Another study reported that approximately 30% of patients had grade 3 or 4 AEs while undergoing treatment with LEN. They additionally reported that thyroid dysfunction and appetite loss during the administration of LEN were independent factors associated with shorter progression-free survival (PFS) [20]. We therefore considered that DLSAE was associated with poor prognosis in patients with HCC treated by LEN.

In contrast, it has been reported that specific AEs caused by MTAs, such as hand-foot-skin reaction (HFSR) and hypertension, are markers that have the potential to predict improvements in therapeutic responses and OS in patients treated with SORA [21,22,23]. However, it is still unclear whether the occurrence of AEs due to LEN, another 1st line MTA, similarly correlates with prognosis. 

The aim of the present study was to investigate the clinical significance of AEs in the prediction of OS in patients with unresectable HCC treated with LEN. As well, the clinical profiles associated with DLSAEs were investigated using a data-mining analysis.

## 2. Results

### 2.1. Patient Characteristics

Patient profiles are summarized in Table 1. The median age was 74 (38–90 years) years old and 19.8% (35/177) of the patients were female. The median body mass index (BMI) was 22.5 kg/m^2^ (15–38.9 kg/m^2^). The etiology of HCC was unrelated to hepatitis B or hepatitis C virus in 34.5 % (61/177) of patients and all patients were classified as Child–Pugh A. ALBI grade 1 was observed in 41.2% (73/177) of patients. The median tumor size was 32 mm (10–127 mm), and 40.7% of patients (72/177) were classified as BCLC stage C. Median alpha-fetoprotein (AFP) levels were 51.2 ng/mL (1.0–146,260 ng/mL) and des-gamma-carboxy prothrombin (DCP) levels were 233.5 mAU/mL (3.3–524,068 mAU/mL). Six patients (8.6%) had a history of receiving treatment with MTAs prior to treatment with LEN. The median observation period was 12.2 months (2.1–29.2 months) (Table 1).

### 2.2. Therapeutic Efficacy of LEN

The therapeutic responses to LEN are shown in Table 2. A complete response (CR) was observed in 3% (6/177) of patients, while 32% (57/177) had a partial response (PR), 41% (72/177) had stable disease (SD), and 24% (42/177) of patients had progressive disease (PD). The overall objective response rate (ORR) was 38% (63/1177) and the disease control rate (DCR) was 76% (135/177) (Table 2).

### 2.3. Type and Timing of AEs

The AEs observed during the course of treatment with LEN are shown in Table 3. The overall incidence of any grade of AE was 83% and the incidence of an AE grade ≥ 3 was 48.9%. Assessed by type of AE, 94 patients (53.1%) experienced hypertension, 93 patients (52.5%) experienced appetite loss, 90 patients (50.8%) experienced fatigue, 86 patients (48.6%) experienced hypothyroidism, 67 patients (37.9%) experienced proteinuria, 50 patients (28.2%) experienced HFSR, and 34 patients (19.2%) experienced diarrhea. The incidence of AEs classified as grade ≥ 3 according to the common terminology criteria for adverse events (CTCAE) was 48.6% (86/177). These included hypertension (14.7%), proteinuria (12.4%), and fatigue (11.3%). The incidence rate of DLSAE was 43.5%. The timing of each AE from the start of LEN administration is shown in Figure 1. The earliest AE to occur was hypertension, and the median onset was only 4 days from the start of LEN (range: 1–366 days).The median timing was 27 days (range: 3–328 days) for the onset of fatigue, 29 days (range: 1–431 days) for the onset of diarrhea, 30 days (range: 5–438 days) for the onset of HFSR, and 31 days (range: 3–359 days) for the onset of appetite loss (Figure 1).

### 2.4. Survival Analysis According to Type of AE

We next assessed whether the type of AE induced by LEN treatment was correlated with OS. Survival curves for each AE are shown in Figure 2. Patients who developed hypertension or HFSR had significantly longer survival compared to those who did not experience these AEs (hypertension median survival time (MST): not reached vs. 14.4 months, *p* = 0.01; HFSR MST: not reached vs. 15.4 months, *p* = 0.04) (Figure 2A,B). In contrast, patients who developed appetite loss had significantly reduced survival compared to those who did not (MST: 15.1 months vs. not reached, *p* = 0.01) (Figure 2C). Fatigue, diarrhea, proteinuria, and hypothyroidism did not significantly influence OS (Figure 2D–G).

### 2.5. Survival Analysis with or without Discontinuation of LEN Due to Severe AEs

Kaplan–Meier analysis for OS according to DLSAE is shown in Figure 2. In the group that did not require DLSAE, the MST was not reached, compared to an MST of 12.8 months in the group that did require DLSAE (Figure 3). The MST in the former group was significantly longer than in the latter group (*p* < 0.001, Figure 3).

### 2.6. Decision-Tree Analysis for Discontinuation of LEN Due to Severe AEs

In this study, the rate of DLSAE in all subjects was 44% at the study end date. To determine the clinical profiles associated with DLSAE, a decision tree analysis was performed, and age was identified as the first splitting variable for the rate of DLSAE. Although the DLSAE rate was only 22% in patients aged < 70, the rate was 57% in patients aged ≥ 71 (Figure 4). In this older group, the second splitting variable according to the decision tree analysis was the ALBI grade. In patients with ALBI grade 1 and 2, the DLSAE rate was 37 and 70%, respectively. In this population, fatigue and appetite loss were identified as the next level of splitting variables (Figure 4). In patients aged ≥ 71 who had ALBI grade 2 and appetite loss grade ≥ 2, the DLSAE rate was 87% (Profile 1 in Figure 4). Similarly, patients ≥ 71 who had ALBI grade 1 and fatigue grade ≥ 3 had a DLSAE rate of 83% (Profile 2 in Figure 4).

### 2.7. Difference in Adverse Events Associated with LEN between the <71 Years and ≥71 Years Groups

Difference in AEs associated with LEN between the < 71 years and ≥ 71 years groups are shown in Table 4. The prevalence of appetite loss and proteinuria were significantly higher in the ≥ 71 years group compared to in the < 71 years group (*p* = 0.01, *p* = 0.01, respectively). In addition, the prevalence of appetite loss grade ≥2 and fatigue grade ≥3 were significantly higher in the ≥ 71 years group compared to in the < 71 years group (*p* = 0.01, *p* = 0.04, respectively).

### 2.8. Logistic Regression Analysis for Discontinuation of LEN Due to Severe AEs

By a stepwise procedure we selected age, ALBI grade 2, fatigue grade ≥ 3, and appetite loss ≥ 2 as the variables in a logistic regression analysis. In this analysis, all 4 of these variables were identified as independent factors for DLSAE (Table 5).

## 3. Discussion

In this study, we demonstrated that advanced age, ALBI grade 2, fatigue grade ≥ 3, and appetite loss grade ≥ 2 were independently associated with DLSAE in patients with HCC treated with LEN. Furthermore, we revealed that hypertension, HFSR, and appetite loss were important predictive factors for the duration of survival in patients with HCC treated with LEN.

This study demonstrated an ORR of 38% and a DCR of 76%. Furthermore, the incidence of any grade AE was 82.1%, while AEs grade ≥ 3 occurred in 49.6% of patients with HCC treated with LEN. These findings are similar to those of the REFLECT study, which previously reported an ORR of 40.6%, a DCR of 73.8%, and severe AEs (grade ≥ 3) of 57% in patients with unresectable HCC treated with LEN [9]. Thus, the therapeutic effect and incidence rate of AEs in the present study appears to be consistent with those that have been previously reported [9,24], suggesting that the patient population, treatment effects and LEN administration protocols in our study were standard.

In this study, patients who developed hypertension and HFSR during LEN treatment survived significantly longer compared to those who did not develop these AEs. In contrast, appetite loss was a poor prognostic factor for survival. Previous studies have shown that patients who experienced hypertension and HFSR while undergoing treatment with SORA had significantly better outcomes than those who did not develop these AEs [25,26,27,28,29]. Our findings indicate that hypertension and HFSR could be also be positive predictive markers of prognosis for patients undergoing treatment with LEN. It remains unclear why hypertension and HFSR could represent markers of improved prognosis. However, these AEs are relatively manageable using supportive drugs, such as antihypertensive drugs and steroids, meaning that it is possible to maintain effective blood concentrations of LEN. The presence of these AEs suggests that the concentration of LEN in the blood inhibited vasculature in the body. In this study, angiotensin receptor blockers (ARB) were administered [30] as an initial antihypertensive drug. Tamaki et al. reported that administration of ARBs had inhibitory effects on HCC and prevented the development of HCC through inhibition of hypoxia-inducible factor-1α (HIF-1α) and the VEGF signaling pathway [31]. Furthermore, in the present study the median onset of hypertension from the start of LEN treatment was only 4 days. These results suggest that the appearance of hypertension has potential as an early surrogate marker to predict longer survival in patients treated with LEN.

Appetite loss is one of the common AEs which leads to discontinuation of LEN treatment. Hiraoka et al. previously reported that appetite loss was associated with the length of time to discontinuation of treatment with LEN [11]. In fact, this study revealed that deterioration of the ALBI score, especially at the 3 months assessment following initiation of LEN, was significantly higher in patients who developed appetite loss as a result of LEN treatment (Appendix A: Figure A1). The ALBI score is calculated using albumin and bilirubin, which are directly correlated with low nutrient condition caused by appetite loss. These may have contributed to the poor prognosis associated with loss of appetite while undergoing LEN treatment.

In this study, the development of other AEs, such as fatigue, diarrhea, hypothyroidism, and proteinuria, during LEN administration had no effect on OS. Although fatigue was not a predictive factor for OS in LEN treatment, 25% of patients who recovered from temporary grade ≥3 fatigue refused re-initiation of LEN. Therefore, grade ≥ 3 fatigue was an independent factor associated with DLSAE. Additionally, patients who developed grade ≥3 fatigue showed significantly worse prognosis (MST: 11.4 months vs. 21.7 months, *p* = 0.01) (Appendix A: Figure A2). Thus, development of fatigue grade ≥3 is also thought to be a critical AE in LEN treatment. We recently reported a useful protocol for LEN involving a 5 days-on/ 2 days-off administration schedule (the weekends-off protocol) [18]. The weekends-off protocol for LEN significantly contributed to improvements in the therapeutic response and tolerance of AEs induced by LEN, thereby prolonging the administration period of LEN and the OS of patients treated with LEN. These findings demonstrate that, to avoid discontinuation of LEN due to AEs, refinement of the administration protocol is important.

The initial split for DLSAE was age ≥ 71 years by the decision tree analysis in this study. In the REFLECT study, only 12.0% of patients were ≥ 75 years old and therefore, an association between advanced age and DLSAE remains unclear [9]. While, in our study, the prevalence of patients with ≥ 75 years old were 48.0%. Advanced aged patients are especially vulnerable to over-treatment (high likelihood of complications/toxicity) [32,33], and there is less evidence to guide chemotherapy treatment decisions for advanced aged patients [34]. Inomata et al. reported that appetite loss is a major adverse event of EGFR-TKIs in elderly patients with non-small cell lung cancer [35]. In addition, aging is associated with higher LEN area under the plasma concentration-time curve (AUC), which result in earlier drug withdrawal or dose reduction [36]. Thus, advanced age may be the most important factor associated with DLSAE.

In this study, advanced age, ALBI grade 2, fatigue grade ≥3, and appetite loss ≥ 2 were identified as independent factors for DLSAEs using multivariate analysis. Yao et al. previously reported that discontinuation of SORA treatment should be prevented to avoid disease progression in HCC [37]. PROSAH (Prediction Of Survival in Advanced Sorafenib-treated HCC) model is also originally designed for evaluation of overall survival of HCC patients treated with sorafenib [38]. In this study, 71 patients (40.1%), 71 patients (40.1%), 30 patients (16.9%), and 5 patients (2.9%) were classified into the risk group 1 (Low), risk group 2 (Intermediate-Low), risk group 3 (Intermediate-High), and risk group 4 (High), respectively. The prevalence of DLSAE was significantly lower in the risk group 1 compared to risk group 2, 3, and 4 (*p* = 0.03). Further, according to analyzed PROSASH model, MST in the risk group 1 was significant longer than other the groups (Appendix A: Figure A3). Moreover, previous studies have reported that ALBI grade 2, fatigue grade ≥ 3, and appetite loss ≥ 2 were predictive factors for therapeutic effect or OS in patients treated by MTAs, including LEN [18,39,40,41]. OS in the group that did not require DLSAE was significantly longer than that of the group that did require DLSAE, indicating that our results were consistent with previous findings [18,37,39,40,41]. Furthermore, Yang et al. previously reported that discontinuation of anti-VEGF therapy promoted metastasis through a liver revascularization mechanism [19]. DLSAE requires a lengthy period to recover from the AEs, and this could induce tumor progression and metastasis. Therefore, the management of MTA-induced AEs is crucial and necessary to avoid disruption of treatment.

The present study had several limitations. First, the study design was retrospective. Second, all patients were all Japanese in this study, and could not focused on the effects of sex, diet, and countries of patients on the DLSAE. Third, we did not evaluate body composition, which has been identified as a prognostic biomarker for treatment-related toxicity and poor survival in HCC [42,43,44]. Fourth, we did not evaluate patient histories of any treatments received previous and/or subsequent to administration of LEN. In this study, 22% (41/177) of patients were previously treated with SORA and REGO. Although LEN treatment was started after the effect of the previous treatment had disappeared, we cannot deny the possibility that the history of the previous treatment might influence the occurrence of AEs during LEN treatment. Moreover, OS is affected by additional therapies after first-line treatment. Generally, sequential therapies should be considered following ineffective treatment for HCC [45,46]. Therefore, to prove the clinical significance of AEs in LEN treatment, it would be necessary to perform a prospective, randomized controlled trial study. Moreover, establishment of a comprehensive grading system to predict DLSAE will be needed in management of LEN treatment.

## 4. Materials and Methods

### 4.1. Study Design and Patients

This retrospective study evaluated 193 patients with unresectable HCC who were treated with LEN between 24 March 2018 and 31 Mar 2020 across the following 5 institutions: Kurume University Hospital (Kurume, Japan), Yokokura Hospital (Miyama, Japan), Omuta City Hospital (Omuta, Japan), Yanagawa Hospital (Yanagawa, Japan) and Iwamoto Internal Medical Clinic (Kitakyushu, Japan). The cut-off date for this analysis was 30 April 2020. Following initial evaluation, 16 patients were excluded and a final total of 177 patients were enrolled in the study (Appendix A: Figure A4).

This protocol conformed to the ethical guidelines of the 1975 Declaration of Helsinki and received approval from the ethics committee of Kurume University School of Medicine (approval number: 18146). Written informed consent for LEN treatment was obtained from each patient. An opt-out approach was used to obtain informed consent from the patients and personal information was protected during data collection.

### 4.2. Evaluation of Hepatic Reserve Function

Liver function was evaluated by the ALBI score and the Child-Pugh score [47]. The ALBI score was calculated as previously described [48] based on serum albumin and total bilirubin levels; ALBI-score = [log10 bilirubin (µmol/L) × 0.66] + [albumin (g/L) × −0.085], and was graded as follows: ≤−2.60 = grade 1, >−2.60 to ≤−1.39 = grade 2, >−1.39 = grade 3).

### 4.3. Treatment Protocol

LEN (Eisai Co., Ltd., Tokyo, Japan) was administered orally to patients with unresectable HCC. The standard dose of LEN therapy was determined based on body weight and liver function, according to the manufacturers’ instruction. LEN was orally administered at a dose of 12 mg/day in patients with body weight ≥ 60 kg, or 8 mg/day in patients with body weight < 60 kg and was discontinued when any unacceptable or severe AEs were observed.

### 4.4. Evaluation of the Therapeutic Response and the Follow-Up Schedule

The therapeutic response was evaluated by dynamic computed tomography (CT) or magnetic resonance imaging (MRI) 4–6 weeks after the initiation of treatment with LEN, according to the Modified Response Evaluation Criteria in Solid Tumors (mRECIST) guidelines [49], and at intervals of 2–3 months thereafter until death or study cessation.

### 4.5. Safety Evaluation and Assessment of Adverse Events

AEs were assessed based on the National Cancer Institute CTCAE, version 4.0 [50]. The dose of LEN was reduced, or treatment interrupted when any AEs of grade 3 or higher or any unacceptable drug-related AEs of grade 2 occurred, in accordance with the administration guidelines. Additionally, we investigated the timing of AEs from the start of LEN treatment and the relationship between AEs and the change in ALBI score after treatment initiation. We also assessed alterations in the patients’ physical condition by telephone interviews.

### 4.6. Statistical Analysis

All data are presented as the number or median (range). All statistical analyses were carried out using statistical analysis software (JMP Pro version 14, SAS Institute Inc., Cary, NC). OS was calculated using the Kaplan-Meier method and analyzed using the log-rank test or Bonferroni method. Between group comparisons were performed using chi-squared test or the Wilcoxon rank-sum test. We also performed a decision tree analysis to identify factors associated with discontinuation due to AEs, as previously described [51]. To select the factors for multivariate analysis, a stepwise procedure was employed as previously described [52,53]. A two-tailed *p* -value of < 0.05 was considered statistically significant.

## 5. Conclusions

In this study, we demonstrated that advanced age, ALBI grade 2, fatigue grade ≥3, and appetite loss ≥ 2 were independently associated with discontinuation of LEN due to AEs in patients with HCC. Moreover, hypertension and HSFR were predictive factors for improved OS. Hypertension in particular may represent a promising early surrogate marker in LEN treatment. Our findings additionally indicate that discontinuation of LEN treatment should be prevented through improved clinical management of AEs for longer survival of patients with HCC.

## Figures and Tables

**Figure 1 cancers-12-01867-f001:**
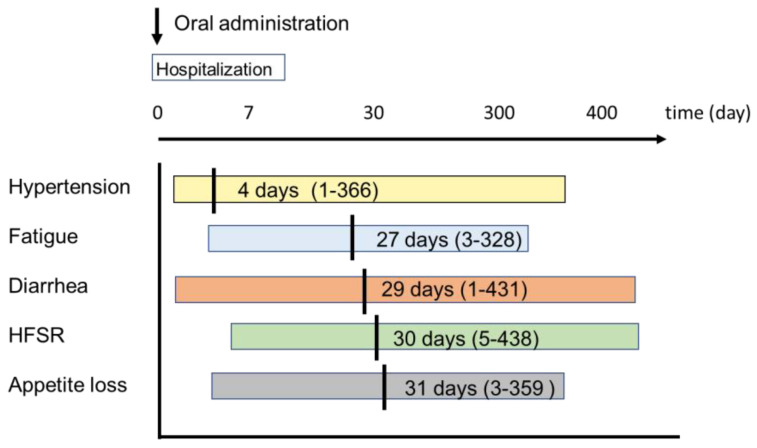
Timing of the occurrence of AEs from initial administration of LEN. All data are expressed as median (range). Yellow line indicates hypertension. Blue line indicates fatigue. Orange line indicates diarrhea. Green line indicates HFSR. Gray line indicates appetite loss. Abbreviation: AE, adverse event; LEN, lenvatinib; HFSR, hand-foot-skin reaction.

**Figure 2 cancers-12-01867-f002:**
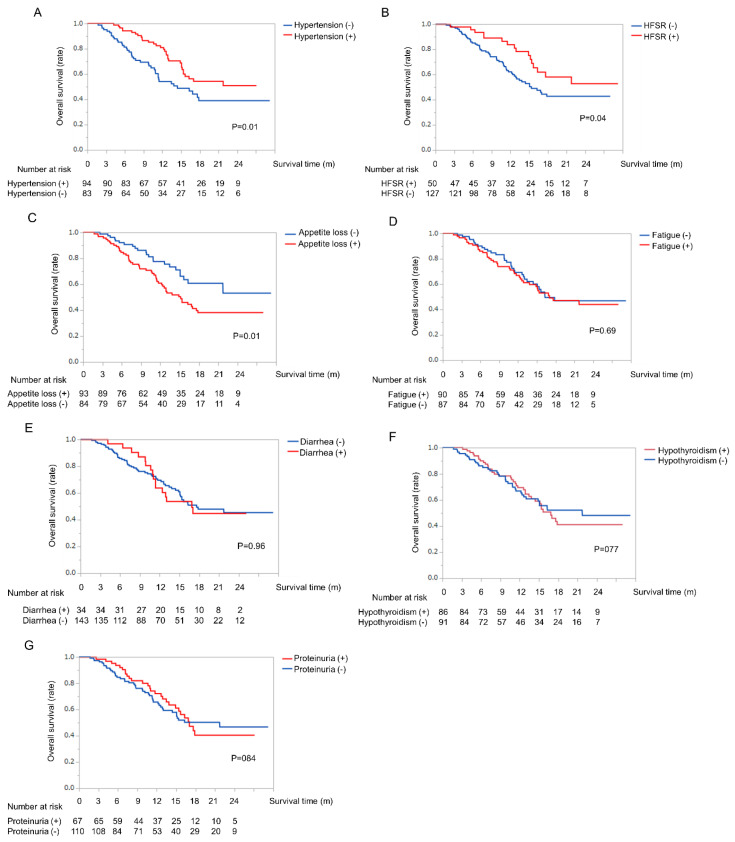
Overall survival time in patients undergoing LEN treatment by type of AE. (**A**) Kaplan-Meier curves for overall survival with and without hypertension. The red line indicates hypertension. The blue line indicates no hypertension. (**B**) Kaplan-Meier curves for overall survival according with and without HFSR. The red line indicates HFSR. The blue line indicates no HFSR. (**C**) Kaplan-Meier curves for overall survival with and without appetite loss. The red line indicates appetite loss. The blue line indicates no appetite loss. (**D**) Kaplan-Meier curves for overall survival with and without fatigue. The red line indicates fatigue. The blue line indicates no fatigue. (**E**) Kaplan-Meier curves for overall survival with and without diarrhea. The red line indicates diarrhea. The blue line indicates no diarrhea. (**F**) Kaplan-Meier curves for overall survival with and without hypothyroidism. The red line indicates hypothyroidism. The blue line indicates no hypothyroidism. (**G**) Kaplan-Meier curves for overall survival with and without proteinuria. The red line indicates proteinuria. The blue line indicates no proteinuria. Abbreviation: HCC, hepatocellular carcinoma; LEN, lenvatinib; AE, adverse events; HFSR, hand-foot-skin reaction.

**Figure 3 cancers-12-01867-f003:**
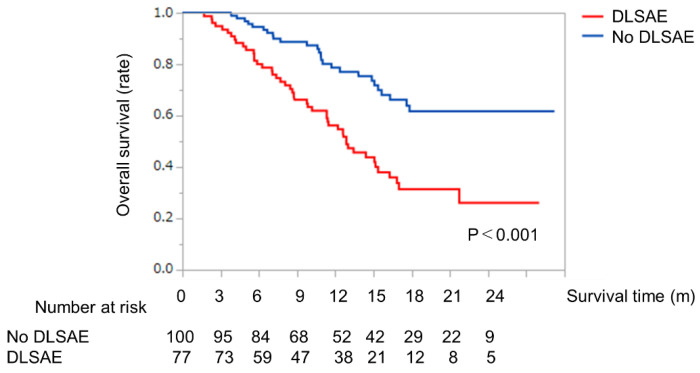
Overall survival time in patients with HCC who did or did not require DLSAE. Kaplan-Meier curves for overall survival according to DLSAE or no DLSAE in patients with HCC treated with LEN. The solid line indicates the no DLSAE group. The dotted line indicates the DLSAE group. Abbreviations: HCC, hepatocellular carcinoma; DLSAE, discontinuation of lenvatinib due to severe adverse events; LEN, lenvatinib.

**Figure 4 cancers-12-01867-f004:**
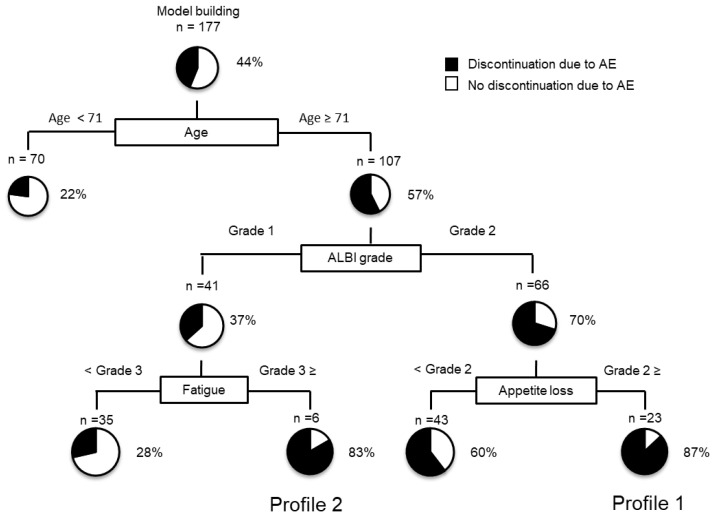
Profiles associated with DLSAE in patients with HCC treated with LEN. Decision-tree algorithm for DLSAE. The pie graphs indicate the percentage of no DLSAE (white)/ DLSAEs (black) in each group. Abbreviation: DLSAE, discontinuation of lenvatinib due to severe adverse events; HCC, hepatocellular carcinoma; LEN, lenvatinib.

**Table 1 cancers-12-01867-t001:** Characteristics of all patients (*n* = 177).

Characteristic	All Patients
N	177
Age (years)	74 (38–90)
Sex (female/male)	35/142
BMI	22.5 (15–38.9)
Etiology (HBV/HCV/Others)	34/82/61
Child-Pugh score (5/6)	135/42
ALBI grade (1/2)	73/104
AST (U/L)	37 (13–160)
ALT (U/L)	29 (6–126)
Diabetes mellitus (+/-)	(75/102)
Maximum tumor diameter (mm)	32 (10–127)
Number of tumors	
<5/≥5	54/123
BCLC stage (B/C)	105/72
AFP (ng/mL)	51.2 (1.0–146,260)
DCP (mAU/mL)	233.5 (3.3–524,068)
Prior treatment of MTAs	41
(SORA/SORA+REGO)	(29/12)
Follow-up duration (month)	12.2 (2.1–29.2)

Note. Data are expressed as median (range), or number. Abbreviations: BMI, Body Mass Index; HBV, hepatitis B virus; HCV, hepatis C virus; ALBI grade, Albumin-bilirubin grade; AST; aspartate transaminase, ALT; alanine aminotransferase, BCLC, Barcelona Clinic Liver Cancer; AFP, α-fetoprotein; DCP, des-γ-carboxy prothrombin, MTAs; molecular target drugs, SORA; sorafenib, REGO; regorafenib.

**Table 2 cancers-12-01867-t002:** Treatment response rate of Lenvatinib (LEN).

Response Category	Patients with HCC Treated with Lenvatinib (*n* = 177)
CR	6 (3%)
PR	57 (32%)
SD	72 (41%)
PD	42 (24%)
ORR	63 (38%)
DCR	135 (76%)

Data are expressed as frequency (percentage). Abbreviations: HCC, hepatocellular carcinoma; CR, complete response; PR, partial response; SD, stable disease; PD, progressive disease; ORR, objective response rate; DCR, disease control rate.

**Table 3 cancers-12-01867-t003:** Adverse events associated with LEN (*n* = 177).

Adverse Event	Any *n* (%)	Grade 1 *n* (%)	Grade 2 *n* (%)	Grade 3 ≥, *n* (%)
Hypertension	94 (53.1%)	12 (6.8%)	56 (31.6%)	26 (14.7%)
Appetite loss	93 (52.5%)	40 (22.6%)	43 (24.2%)	10 (5.6%)
Fatigue	90 (50.8%)	24 (13.6%)	46 (25.9%)	20 (11.3%)
Hypothyroidism	86 (48.6%)	28 (15.8%)	58 (31.0%)	0 (0%
Proteinuria	67 (37.9%)	27 (15.6%)	18 (10.2%)	22 (12.4%)
HFSR	50 (28.2%)	22 (12.7%)	17 (9.6%)	11 (6.2%)
Diarrhea	34 (19.2%)	11 (6.2%)	11 (6.2%)	12 (6.8%)
Liver disorder	49 (27.7%)	39 (22.0%)	6 (3.3%)	4 (2.3%)
Hoarseness	45 (25.4%)	42 (23.7%)	3 (1.7%)	0 (0%)
Thrombocytopenia	34 (19.2%)	4 (2.3%)	21 (11.9%)	9 (5.0%)
Hepatic encephalopathy	18 (10.1%)	2 (1.1%)	15 (8.4%)	1 (0.5%)
Ascites	7 (3.9%)	5 (2.8%)	2(1.1%)	0 (0%)
Drug-induced pneumoniae	3 (1.7%)	0 (%)	3 (1.7%)	0.0 (0)

LEN, lenvatinib; HFSR, hand-foot-skin-reaction.

**Table 4 cancers-12-01867-t004:** Difference in adverse events associated with LEN between the <71 years and ≥ 71 years groups.

Characteristic	All Patients	Age < 71	Age ≥ 71	*p*
*n*	177	70	107	
Hypertension (Presence/Absence)	94 (53.1%)	34/36 (48.6%/51.4%)	60/47 (56.0%/44.0%)	0.32
Appetite loss (Presence/Absence)	93 (52.5%)	29/41 (41.4%/58.6%)	64/43 (59.8%/40.2%)	0.01
Fatigue (Presence/Absence)	90 (50.8%)	35/35 (50.0%/50.0%)	55/52 (51.4%/48.6%)	0.86
Hypothyroidism (Presence/Absence)	86 (48.6%)	31/39 (44.3%/55.7%)	55/52 (51.4%/48.6%)	0.35
Proteinuria (Presence/Absence)	67 (37.9%)	19/51 (27.1%/72.9%)	48/59 (44.9%/55.1%)	0.01
HFSR (Presence/Absence)	50 (28.2%)	25/45 (35.7%/64.3%)	25/82 (23.4%/76.6%)	0.08
Diarrhea (Presence/Absence)	34 (19.2%)	14/56 (20.0%/80.0%)	20/87 (18.7%/81.3%)	0.82
Appetite loss grade ≥2 (Presence/Absence)	53 (30.8%)	14/56 (20.0%/80.0%)	39/68 (36.4%/63.6%)	0.01
Fatigue grade ≥3 (Presence/Absence)	20 (11.3%)	4/66 (5.7%/94.3%)	16/91 (14.9%/84.1%)	0.04

Abbreviations: LEN, lenvatinib; HFSR, hand-foot-skin-reaction.

**Table 5 cancers-12-01867-t005:** Multivariate analysis factors which associated with DLSAEs.

Factors	Unit	Odds Ratio	95% Confidence Interval	*p*
Age	1	1.12	1.04–1.12	<0.001
ALBI grade 2	N/A	3.15	1.54–6.69	0.001
Fatigue grade ≥3	N/A	4.49	1.34–18.4	0.008
Appetite loss grade ≥2	N/A	2.69	1.24–5.97	0.012

Abbreviations: DLSAE; discontinuation of lenvatinib due to severe adverse events, ALBI; albumin-bilirubin.

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
