# Peer review of "Clinical Significance of Adverse Events for Patients with Unresectable Hepatocellular Carcinoma Treated with Lenvatinib: A Multicenter Retrospective Study"

_cancers, 2020, doi:10.3390/cancers12071867_

Round 1

Reviewer 1 Report

The authors have already addressed all of the concerns and explained the rational of the decision tree. This is agreed that this results from this study is a first-hand information to support that adverse effects can be factors for making the decision whether LEN should be continues or not.  

Some minor comments

1.       In figure 1, It will be good to indicate if the number shown in the figure is mean/mode/median in the legend.

2.       In table 4, it will be good to include the statistical analysis used to obtain the P value in the figure legend or in Methodology section.

3.       “Table4. Difference in adverse events associated with LEN between the <71 years and ≥ 71 years groups” and “Table 4. Multivariate analysis factors which associated with DLSAEs.” Please correct the table number in the manuscript.

Author Response

To REVIEWER 1

Thank you very much for your letter regarding our manuscript (cancers-857700). We appreciate your comments, which have helped us to improve our manuscript. In line with your comments, please find below our point-by-point responses.

1) In figure 1, It will be good to indicate if the number shown in the figure is mean/mode/median in the legend.

Answer: We appreciate your comment. We apologize for unclear description in the Figure 1. In the Figure 1, all data are expressed as median (range). Following your suggestion, we indicated it in the Figure 1 legend in the revised manuscript (Line 126-127). Again, we appreciate your valuable comment, which have helped us to improve our manuscript.

2) In table 4, it will be good to include the statistical analysis used to obtain the P value in the figure legend or in Methodology section.

Answer: As you indicated, we apologize that we did not describe the statistical analysis for table 4. We have added the following description in the methodology section of the revised manuscript: “Between group comparisons were performed using chi-squared test or the Wilcoxon rank-sum test.” (Line 331-332).

3) “Table4. Difference in adverse events associated with LEN between the <71 years and ≥ 71 years groups” and “Table 4. Multivariate analysis factors which associated with DLSAEs.” Please correct the table number in the manuscript.

Answer: We appreciate your careful proof reading. Following your suggestion, we revised the description. (Line 194, 196)

Reviewer 2 Report

Unresectable hepatocellular carcinoma remains very challenging. I think this revised version describing a high-quality retrospective study on the factors provoking disconituation of lenvatinib therapy and especially the identification of hypertension as a sign of good prognosis is a very valuable contribution to the field. I recommend publication of manuscript as is.

Author Response

To REVIEWER 2

Thank you very much for your letter regarding our manuscript (cancers-857700). We appreciate your comments which, have helped us to improve our manuscript.

1) Unresectable hepatocellular carcinoma remains very challenging. I think this revised version describing a high-quality retrospective study on the factors provoking disconituation of lenvatinib therapy and especially the identification of hypertension as a sign of good prognosis is a very valuable contribution to the field. I recommend publication of manuscript as is.

We are very glad to hear that the reviewer understand the importance of our study. Again, we appreciate reviewer’s valuable comment, which have helped us to improve our manuscript.

This manuscript is a resubmission of an earlier submission. The following is a list of the peer review reports and author responses from that submission.

Round 1

Reviewer 1 Report

In this manuscript, the authors mentioned that the appearance of side effect from lenvatinib (LEN) could be used as markers for predicting the prognosis. The study is important to provide suggestions for determining whether LEN treatment should be discontinued. However, the results from this manuscript is largely overlapped with the results from REFLECT study which also included Japanese patients. Therefore, the novelty is significantly compromised.

In this manuscript, the authors did mention one important thing that was to establish a grading system based on the side effect to determine if LEN should be stopped (Figure 4). As the clinical management for LEN treatment has been establishing and improving, the authors did provide an effort for making the management process. It is appreciated.

I would suggest the authors to build a comprehensive grading system rather than putting their effort on repeating something that was studied. In figure 4, it is not understand why age would be the first step to decide if the patient should continue LEN treatment. The sequence of using which of the side effects to make the decision was not clear. The authors should work on this to build a more robust model.    

Reviewer 2 Report

The reviewer want to know whether the patients are Japanese or patients globally. Will the results different if the diet, sex, countries of the patients are different ? Is it possible to know more about the methodology ?

Reviewer 3 Report

This is a short but well-prepared contribution. I have only a few questions/suggestions

Introduction. First section. To improve attraction of the manuscript and place it into the context of recent developments it may be worthwhile to mention that prediction of HCC risk is improving, see e.g. Papatheodoridis GV, Sypsa V, Dalekos GN, Yurdaydin C, Van Boemmel F, Buti M, Calleja JL, Chi H, Goulis J, Manolakopoulos S, Loglio A, Voulgaris T, Gatselis N, Keskin O, Veelken R, Lopez-Gomez M, Hansen BE, Savvidou S, Kourikou A, Vlachogiannakos J, Galanis K, Idilman R, Esteban R, Janssen HLA, Berg T, Lampertico P. Hepatocellular carcinoma prediction beyond year 5 of oral therapy in a large cohort of Caucasian patients with chronic hepatitis B. J Hepatol. 2020 Jun;72(6):1088-1096.

For Sorafinib body composition is an important factor: Labeur TA, van Vugt JLA, Ten Cate DWG, Takkenberg RB, IJzermans JNM, Groot Koerkamp B, de Man RA, van Delden OM, Eskens FALM, Klümpen HJ. Body Composition Is an Independent Predictor of Outcome in Patients with Hepatocellular Carcinoma Treated with Sorafenib. Liver Cancer. 2019 Jul;8(4):255-270. What about Lenvatinib. Have the authors data on this? Similarly, how do variables driving performance is in the Prosash models relate to the effects seen in the present study ? ( Labeur TA, Berhane S, Edeline J, Blanc JF, Bettinger D, Meyer T, Van Vugt JLA, Ten Cate DWG, De Man RA, Eskens FALM, Cucchetti A, Bonnett LJ, Van Delden OM, Klümpen HJ, Takkenberg RB, Johnson PJ. Improved survival prediction and comparison of prognostic models for patients